# Deconstructing sarcomeric structure–function relations in titin-BioID knock-in mice

Franziska Rudolph[1], Claudia Fink[1], Judith Hüttemeister[1], Marieluise Kirchner[2], Michael H. Radke [1,3], Jacobo Lopez Carballo[1], Eva Wagner[4,5,6], Tobias Kohl[4,5,6], Stephan E. Lehnart [4,5,6], Philipp Mertins [2,7] & Michael Gotthardt [1,8✉]

Proximity proteomics has greatly advanced the analysis of native protein complexes and subcellular structures in culture, but has not been amenable to study development and disease in vivo. Here, we have generated a knock-in mouse with the biotin ligase (BioID) inserted at titin's Z-disc region to identify protein networks that connect the sarcomere to signal transduction and metabolism. Our census of the sarcomeric proteome from neonatal to adult heart and quadriceps reveals how perinatal signaling, protein homeostasis and the shift to adult energy metabolism shape the properties of striated muscle cells. Mapping biotinylation sites to sarcomere structures refines our understanding of myofilament dynamics and supports the hypothesis that myosin filaments penetrate Z-discs to dampen contraction. Extending this proof of concept study to BioID fusion proteins generated with Crispr/CAS9 in animal models recapitulating human pathology will facilitate the future analysis of molecular machines and signaling hubs in physiological, pharmacological, and disease context.

---

[1] Neuromuscular and Cardiovascular Cell Biology, Max Delbrück Center for Molecular Medicine in the Helmholtz Association, Robert Rössle Strasse, 1013125 Berlin, Germany. [2] Proteomics Platform, Max Delbrück Center for Molecular Medicine in the Helmholtz Association, Robert Rössle Strasse, 1013125 Berlin, Germany. [3] DZHK (German Center for Cardiovascular Research), Partner Site Berlin, Berlin, Germany. [4] Heart Research Center Göttingen, University Medical Center Göttingen, Göttingen, Germany. [5] Department of Cardiology and Pneumology, University Medical Center Göttingen, Göttingen, Germany. [6] DZHK (German Center for Cardiovascular Research), Partner Site Göttingen, Göttingen, Germany. [7] Berlin Institute of Health (BIH), Berlin, Germany. [8] Charité Universitätsmedizin, Berlin, Berlin, Germany. ✉email: gotthardt@mdc-berlin.de

The myofilament is one of the largest and most intricate macromolecular assemblies in living organisms with almost 200 different sarcomere-associated proteins arranged in a periodic pattern[1,2]. This highly dynamic complex not only includes structural proteins but also signaling molecules and enzymes. It undergoes extensive adaptation and remodeling in development and disease[3–5]. While actin and myosin mainly contribute to the contractile properties of the sarcomere, the giant protein titin determines its elasticity[6]. Titin provides the sarcomeric backbone as it integrates into the Z-disc and M-band where it overlaps with titins of the neighboring half-sarcomere to form a continuous filament along the myofiber[7–9]. How titin is inserted into the sarcomere is still unclear and although multiple titin integration sites have been mapped, its positioning even within the thoroughly characterized Z-disc[8,10,11] has not been fully resolved. This extends to the role of the Z-disc during contraction —specifically at very short sarcomere lengths, where the sliding filament[12,13] and crossbridge theory[14] do not sufficiently explain the amount of force generated as actin and myosin filaments interact. A recent theoretical model that builds on the hypothesis that myosin filaments slide through the Z-disc replicates the force–length relation along the full physiological sarcomere length[15], but awaits experimental validation.

Here we insert the biotin ligase BioID[16] at titin's Z-disc in a knock-in approach to survey the sarcomeric proteome in vivo and link the myofilament to specific signaling and metabolic pathways in neonatal vs. adult heart and skeletal muscle, remap titin at the Z-disc and provide experimental evidence to support the myosin/Z-disc sliding model[15].

## Results

**Localization proteomics to study sarcomere dynamics in vivo.** To probe the Z-disc protein environment, we used homologous recombination in embryonic stem (ES) cells and inserted the biotin ligase BioID into the titin filament (Fig. 1a and Supplementary Fig. 1a–c). Placement at the transition from Z-disc to I-band in Exon 28 was well tolerated—similar to the knock-in of DsRed into the same location[17]. Homozygous TiZ-BioID$^{P/P}$ mice followed the same growth curve as wild-type controls, had normal size hearts, were fertile, produced offspring at the expected Mendelian ratios, expressed the transgene at expected levels and properly inserted the BioID at the Z-disc (Fig. 1c, Supplementary Fig. 1c–l, and Supplementary Table 1). As physiological expression of BioID-titin resulted in significantly smaller amounts of biotinylated peptides compared with previous studies with overexpression of BioID fusion proteins[16,18], we adapted the protocol for improved peptide retrieval using trypsin-digested lysates derived from cryo-fractured tissue powder as an input and an anti-biotin antibody[19,20], which facilitates release of the precipitated peptides for subsequent mass spectrometry (MS) analysis as compared with streptavidin. The most prominently biotinylated protein was titin with the expected hotspot of biotinylated peptides around the BioID insertion site (Fig. 1d, e). We estimated the radius of action at 7–15 nm based on the length of the flanking Ig domains (~40 Å) and unstructured regions (4 Å per amino acid) (Fig. 1e, bottom). This confirms earlier work on the nuclear pore complex with a BioID labeling radius of 10 nm[21]. A secondary hotspot was present at titin's Z-repeats, with similar spatial constraints (Fig. 1e, dashed circles), suggesting backfolding and transposition of Ig8/9 to the edge of the Z-disc. Here, labeling intensities amount to 4% of the primary biotinylation hotspot in quadriceps vs. 7.5% in the heart. If we assume that the amount of biotinylation is proportional to the duration of colocalization, this would suggest juxtaposition of titin's Z-repeats and Ig8/9 over 4–8% of the duration of the cardiac cycle at end-systole, when sarcomere length is minimal.

**Titin's intramolecular interactions from Z-disc to M-band.** Titin biotinylation is confined to specific subregions of the protein, extending all the way into the M-band. Biotinylation sites are strongly conserved in the heart and skeletal muscle with minor differences attributable to differential isoform expression (Fig. 1e, at AA 10,000 in quadriceps and at AA 4000 in the heart). The titin A-band region is not biotinylated, but low-level biotinylation at the I-band between AA 9000 and 13,000 and at the M-band suggest specific long-range interactions that affect a minority of titin proteins. This could apply to nascent, soluble titin proteins that are not yet integrated into the sarcomere. Their additional structural flexibility vs. sarcomeric titin would allow the BioID domain at titin's Z-disc region to approach downstream domains as far as titin's M-band and form a compact titin transport variant with improved mobility. This is consistent with recent data on the titin lifecycle, where localized synthesis at the Z-disc and distribution of soluble titin throughout the cytoplasm precedes integration into the sarcomere lattice[17].

We validated the localization of biotinylated lysines flanking the Z-disc using super-resolution imaging (STED, Fig. 1g) and confirmed the reversible association of Ig8/9 with the Z-disc by staining stretched flexor digitorum brevis muscle (Fig. 1h). At increased sarcomere lengths (+10%), the secondary biotinylation hotspot is localized outside the Z-disc ~200 nm from the edges of the Z-disc, which widens upon stretch (Fig. 1f–h). Soluble titin contributes to the background signal (traces Fig. 1g, h) with relative fluorescence intensities at the C-terminal I-band, secondary and primary biotinylation sites matching the frequencies of the respective biotinylated peptides.

**A titin Z-disc complex with enzymes and signaling molecules.** In addition to biotinylated titin peptides, which amount to >90% of total biotinylated peptides, we identified 15 biotinylated proteins related to muscle filament assembly, myocyte function, α-actinin binding, and metabolism (Fig. 2a and Supplementary Fig. 2a, b). Some of these have independently been identified in the heart and skeletal muscle such as Ldb3, Pgam2, and Phtf1 (Fig. 2b–d). Metabolic enzymes include phosphoglycerate mutase (heart and quadriceps, Fig. 2c) with its role in glycolysis, as well as creatine kinase, which buffers ATP in skeletal muscle (Supplementary Fig. 2c). Among the expected Z-disc-associated proteins are Ldb3, plectin, and the heart-specific nebulette (Fig. 2b and Supplementary Fig. 2d, e). Our identification of the endoplasmic reticulum (ER) protein Phtf1, which may play a role in transcriptional regulation and Speg (Fig. 2d and Supplementary Fig. 2f) supports the link between Z-disc and sarcoplasmic reticulum (SR), where Speg controls calcium reuptake in diastole[22], a process in tune with titin's elastic properties that determine diastolic filling. Spectrins have been linked to disrupted Z-discs[23] and Copg2 is involved in ER–Golgi transport[24]. In the majority of these proteins, biotinylation is limited to a single site outside a functional domain and replicated in the heart and skeletal muscle (Fig. 2b–d and Supplementary Fig. 2c–h), suggesting a specific orientation of the respective complex. Their localization within the sarcomere includes the Z-disc, but can extend far into the A-band—including Myosin as a classical A-band protein (Fig. 2h).

**Myosin filaments penetrate the Z-disc during contraction.** Towards elucidating the intersection of Z-disc and myosin filament, we found myosin light- and heavy chains biotinylated in both heart and skeletal muscle (Fig. 2e–g and Supplementary Fig. 2i, j). Biotinylation of Myh4 and -8 was restricted to multiple largely overlapping sites in skeletal muscle and to the very C terminus of Myh6 in the heart, all within the unstructured myosin rod. Myosin motor domains are devoid of biotinylation,

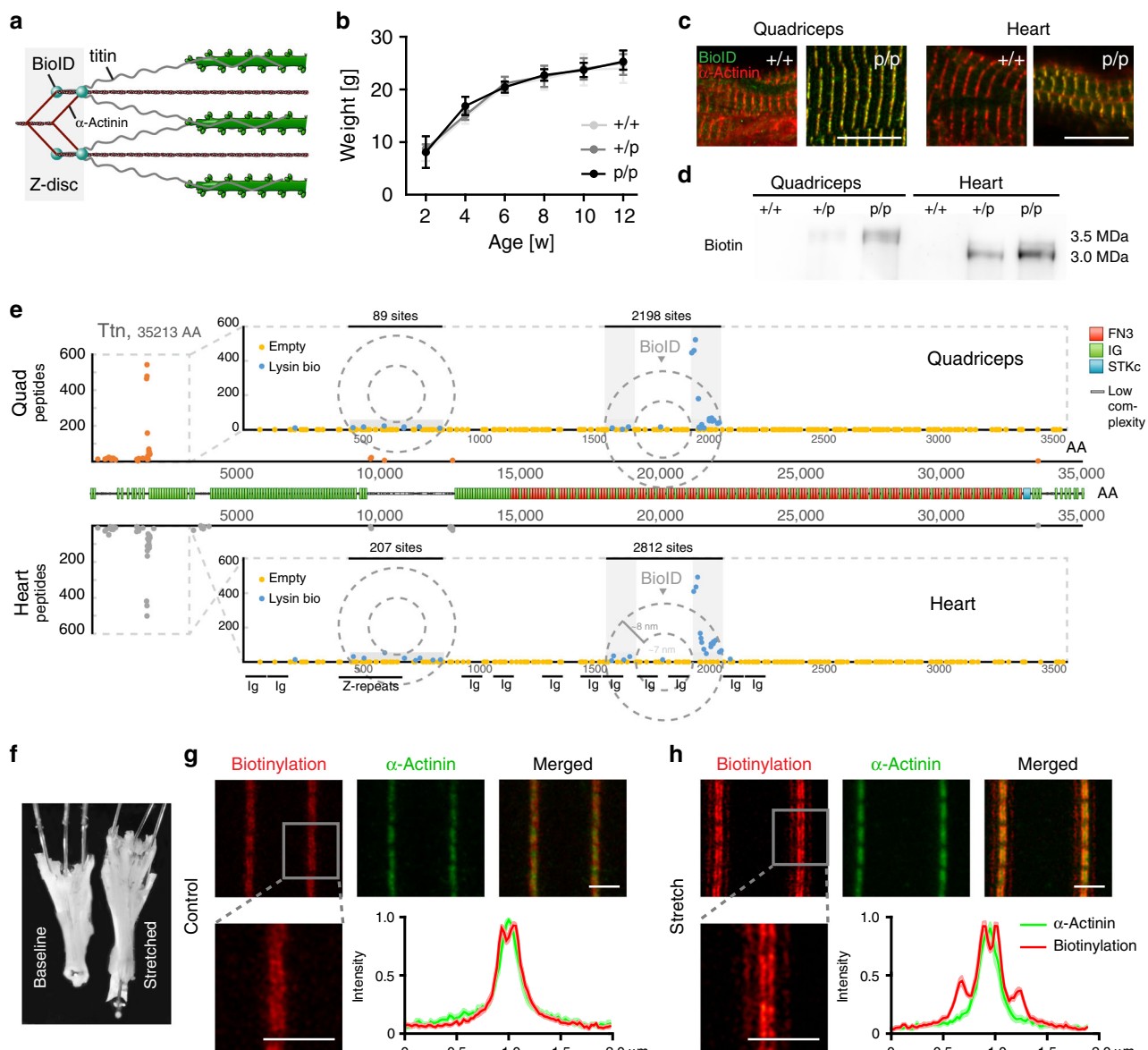

**Fig. 1 Titin Ig8/9 dissociates from the Z-disc upon stretch. a** Knock-in of BioID between Z-disc and I-band titin to probe the Z-disc proteome. **b** Body weight develops normally in TiZ-BioID knock-in- (p/p or p/+) and control mice (+/+). Mean ± SD, n = 23. Two-way ANOVA p < 0.0001 for time, not significant for genotype or interaction. **c** Z-disc colocalization of BioID (green) and α-actinin (red) at the Z-disc. Scale bar 10 μm. **d** Ligand blot with streptavidin with biotinylated titin proteins at the expected sizes and increased signal in homo- vs. heterozygotes. **e** Biotinylation sites along the titin protein. Dashed boxes represent the magnified extended Z-disc region with biotinylation hotspots around the BioID insertion site and the end of the Z-repeats (biotinylated lysin blue, unmodified lysins orange). Dashed circles indicate the range of BioID activity. **f** Flexor digitorum brevis muscle at resting length (baseline) or stretched by 10% and fixed with minutien pins. **g** Stimulated Emission Depletion Microscopy (STED) of flexor digitorum brevis muscle (FDB) without mechanical strain (fixed at slack length) from TiZ-BioID mice (streptavidin staining for biotinylated proteins, red; anti-α-actinin for Z-disc, green). Boxes indicate the area of quantification for the intensity plots of the biotinylation signal and α-actinin (n = 9; mean ± SEM). **h** STED of FDB stretched by ~10%. A biotin hotspot is dislocated from the Z-disc upon stretch. Traces represent mean ± SEM, n = 9. Scale bar 1 μm. Source data are provided as a Source Data file.

which could be related to their movement interfering with access to the BioID and would be consistent with a significant extension of the myosin filament into the Z-disc past the proximal myosin heads. We used super-resolution microscopy to validate the localization of myosin at the edge of the Z-disc in relaxed sarcomeres (Fig. 2h–j) and dislocation from the Z-disc upon stretch (Fig. 2k, l), where the myosin signal is clearly distinct from the secondary titin hotspot (Fig. 2k). The localization of myosin at the Z-disc in contracted sarcomeres (sarcomere length = 1.7 μm, Supplementary Fig. 2n) and the discrete biotinylation of the myosin heavy chain rod support the model of myosin penetrating

the Z-disc to dampen contraction at very short sarcomere lengths as proposed recently[15].

**Developmental vs. adult functions of the titin hub**. To extend our approach of in vivo colocalization proteomics from Z-disc titin to the sarcomere, we used an adapted streptavidin-based pulldown protocol eliminating the initial trypsin digest to enrich entire sarcomere complexes. This protocol identified 478 sarcomere-associated proteins, of which 7 were biotinylated vs. 9 biotinylated proteins in the antibody pulldown (Supplementary

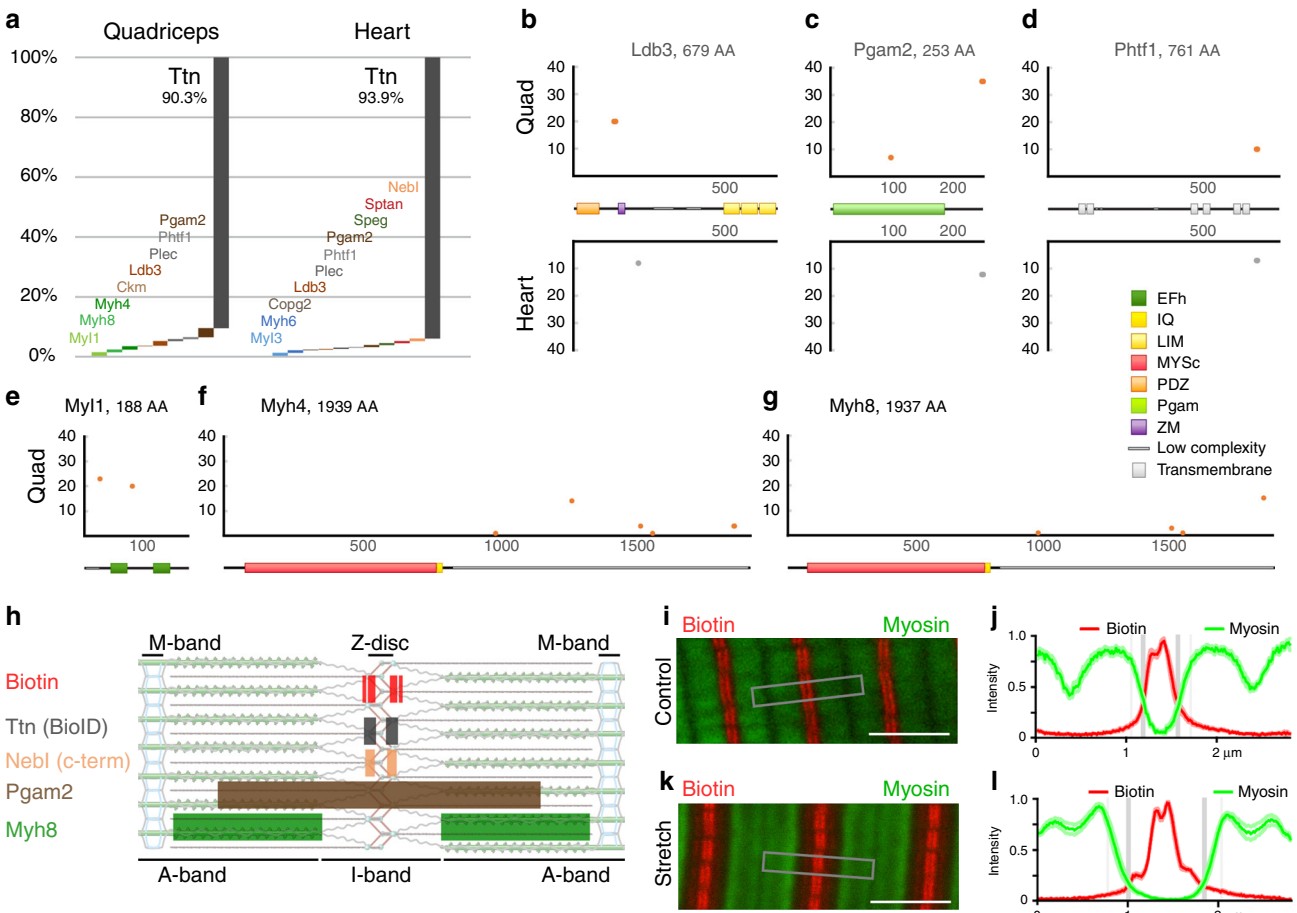

**Fig. 2 Metabolic enzymes and myosin enter the Z-disc in vivo. a** In addition to major biotinylation of titin (>90%) and known Z-disc proteins (Ldb3, Plec, Nebl), several metabolic enzymes and components of the myosin filament are biotinylated in striated muscle in vivo. **b** Ldb3 is differentially spliced in quadriceps and heart resulting in alternative biotinylation of differentially included domains (orange vs. gray dots). Phosphoglycerate mutase (Pgam2, **c**) and putative homeodomain transcription factor 1 (Phtf1, **d**) are unexpected Z-disc-associated proteins with specific and consistent biotinylation between the heart and skeletal muscle. **e–g** Myl1, Myh4, and Myh8 are skeletal muscle specific with biotinylation outside the myosin/actin interface and overlapping biotinylation patterns between Myh4 and Myh8 towards the C terminus. **h** Summary of localization data for biotinylated proteins (Supplementary Fig. 2). **i–l** Super-resolution imaging (STED) of FDB muscle from TiZ-BioID mice at different sarcomere lengths (streptavidin staining for biotinylated proteins, red; anti-myosin, green). **i** Control FDB without mechanical strain. Box indicates one of the nine sample areas used for quantification. **j** Intensity plots indicating the spacing of myosin vs. biotinylation (n = 9; mean ± SEM). Light gray edge of the myosin filament at 75% signal intensity, dark gray area of overlap (both biotin and myosin signal intensity >25%). **k, l** FDB stretched by ~10%. The secondary biotin hotspot dislocated from the Z-disc upon stretch is distant from the myosin signal (no overlap at >25% signal intensity). Traces represent mean ± SEM, n = 9. Scale bar 2 μm. Source data are provided as a Source Data file.

Fig. 3, >90% confirmed by independent replicate). More than 50 of these proteins cover all subregions of the sarcomere from Z-disc to M-band and an additional subset relates to the SR, supporting the link between sarcomere and SR established by antibody pulldown (comparison Fig. 2). The majority of proteins identified do not relate to the core sarcomere structure (Gene Ontology (GO) analysis for cellular component), but most of them have previously been described to interact with sarcomeric proteins (Figs. 3 and 4, and Supplementary Fig. 3). Proteins not previously associated with the sarcomere include collagen (Col5A1). As an extracellular protein, this could be considered a false positive. Alternatively, titins might be released from myocytes undergoing cell death to biotinylate extracellular proteins[25].

Extending our findings to the neonatal sarcomere, we enriched sarcomeric proteins from newborn TiZ-BioID[p/p] mice and identified several proteins, which link the sarcomere scaffold to striated muscle contraction, metabolism, and signal transduction (Fig. 5 and Supplementary Fig. 4). Although all tissues and

developmental stages relate to Striated Muscle Contraction as the dominant WikiPathway term, we find a major shift from proteins regulating gene expression (mRNA processing and cytoplasmic ribosomal proteins) and signal transduction in neonatal striated muscle to metabolism in the adult tissue. Our analysis confirms differences in the adult heart and quadriceps: the heart primarily metabolizes fatty acids or glucose (Fig. 5g; WikiPathway: TCA cycle and fatty acid beta oxidation). Quadriceps consists of a mix of fast and slow fibers metabolizing glucose, fatty acids, and ketone bodies[26] (Fig. 5i: top WikiPathway: glycolysis, oxidative phosphorylation, electron transport chain). Neonatal heart and quadriceps link to proteins associated with mammalian target of rapamycin (mTOR), mitogen-activated protein kinase (MAPK), and nuclear factor-κb (NF-κb) signaling, with only the latter preserved from neonatal to adult skeletal muscle. NF-κb signaling is required for striated muscle hypertrophy and involves translocation of the protein from the cytosol to nucleus[27]. As we find several signaling pathways and nuclear proteins

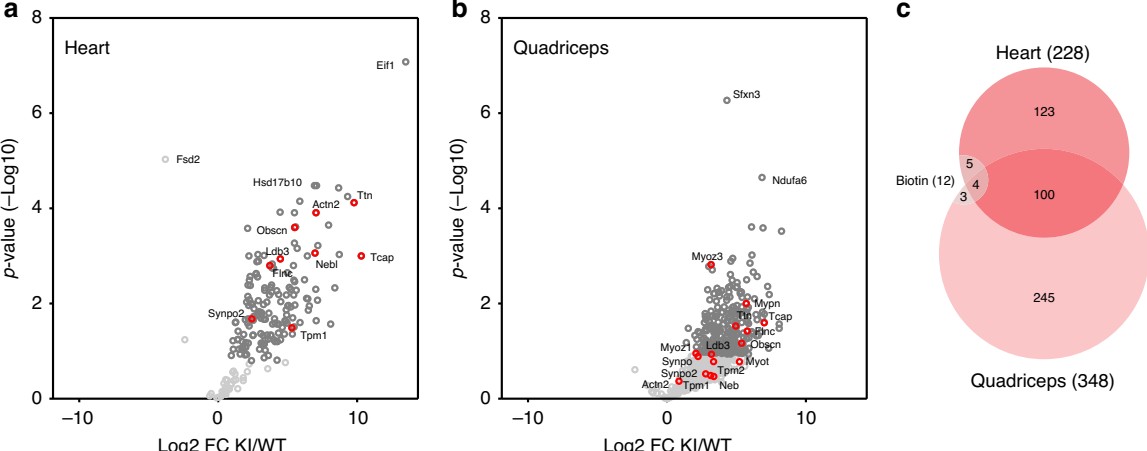

**Fig. 3 The sarcomeric proteome in the heart and skeletal muscle. a–c** Fold change and *p*-value of proteins that co-purify with biotinylated proteins after streptavidin pulldown of TiZ-BioID knock-in vs. WT hearts (**a**) and quadriceps (**b**) with known Z-disc-associated proteins in red, significantly enriched in dark gray. **c** The adult sarcomeric proteome comprises >450 proteins, 100 of them shared between heart and quadriceps. Less than 5% of proteins retrieved were biotinylated. Source data are provided as a Source Data file.

connected to the sarcomere (Fig. 5 and Supplementary Table 2), we suggest that nuclear translocation from the sarcomere is a general principle to intersect sarcomere biomechanics and signal transduction.

## Discussion

Myofilament proteins are at the core of heart and skeletal muscle disease and increasingly recognized as targets for the development of therapeutic strategies[28,29]. Significant progress has been made towards understanding how they interact within the sarcomere lattice and integrating this information into predictive models that describe how changes in active and passive properties of the sarcomere adapt striated muscle function[30]. Despite the wealth of biomechanical, imaging, omics, and structural data, we still struggle to describe the complexity of the sarcomere at the molecular level—in part due to the lack of a comprehensive annotation of sarcomeric proteins, their interactions and their connection to striated muscle signaling, biomechanics, and metabolism.

Here we have generated a titin-BioID knock-in mouse to provide a census of the sarcomeric proteome in heart and skeletal muscle. In adults, we identified 478 sarcomeric or sarcomere-associated proteins—more than doubling the known sarcomeric proteome (the GO database lists 190 mouse proteins connected to the term sarcomere by experimental evidence, inferred from ancestors/sequence homology or by author statements—GO:0030017 http://www.informatics.jax.org). We replicated our findings in two independent experiments each with ~90% overlap and extended the analysis to two developmental stages. As an additional level of validation, we intersected our sarcomeric proteome with published interaction data and found all but 34 proteins as part of an extended network (Supplementary Fig. 3a).

In addition to structural proteins, adaptors, and metabolic enzymes predominantly expressed in adult heart and skeletal muscle, we link neonatal signaling pathways to the sarcomere. These include MAPK, mTOR, and NF-κB signaling, and have previously been associated with cardiomyocyte maturation and growth[31,32]. The shift from sarcomeric signaling and elevated gene expression to increased metabolic activity reflects the transition from assembly and maturation of the neonatal sarcomere to optimized performance, meeting the increased mechanical demands in the adult heart and skeletal muscle. In addition, the association of multiple signaling and metabolic pathways to the sarcomere

reinforces its role in mechanotransduction to balance contractility, growth, and energy production—dependent on load (Fig. 6a).

In-vitro analyses including three-dimensional structures of sarcomeric proteins, interaction studies, polymerization, and functional assays have greatly improved our understanding of the sarcomere. Here we extend those findings in an in-vivo system. As we integrate colocalization information over time with consistently high biotin levels from prenatal through adult, we were able to accumulate sufficient signal to remap Z-disc titin and its immediate protein environment at amino acid resolution. We extend published interaction data between Z-disc titin and α-actinin[8] to suggest strain-dependent backfolding of titin at the Z-disc that could serve both to adapt its elastic properties and as a Z-disc-based stretch sensor (Fig. 6b). Here, the distribution of the biotinylation sites suggests movement of titin's BioID integration site not along α-actin but formation of a hairpin as the sarcomere contracts (Supplementary Fig. 5). This hairpin can contribute to adapting the elastic properties of the sarcomere, which have so far largely been assigned to the N2B, PEVK, proximal and distal Ig domains that act as entropic springs in series[33]. The extended hairpin would change preload on titin's remaining I-band and thus provide an additional level of regulation to adapt its elastic properties. It would thus extend the worm-like chain model[33] with I-band titin composed of the hairpin, the elastic spring domains, and spacer regions arranged in series. This added flexibility would provide adapted mechanical properties along a wider range of sarcomere lengths.

The relative intensities of biotinylation around the BioID compared with the Z-disc core (proximal to the Z-repeats) can be used to derive the timing of sarcomere contraction. In the fully contracted state, titin's Z-repeats and Ig8/9 are juxtaposed to enable biotinylation at the Z-disc core. Here, biotinylation at 4–8% vs. the BioID integration site as a reference in skeletal muscle and heart would be consistent with a delayed start of relaxation that lasts ~0.01 s of the murine ~0.15 s heartbeat. We also find autobiotinylation of titin along the full-length protein, which implies the existence of a compacted, soluble titin to facilitate the previously documented movement of the giant protein along and between the myofilament lattice[34]. Biotinylation at multiple discrete sites suggests a limited number of specific conformations of the soluble titin proteins and the low biotinylation intensities towards the I- and M-band are consistent with the proposal that only a minor fraction of the total titin pool is not integrated into the sarcomere.

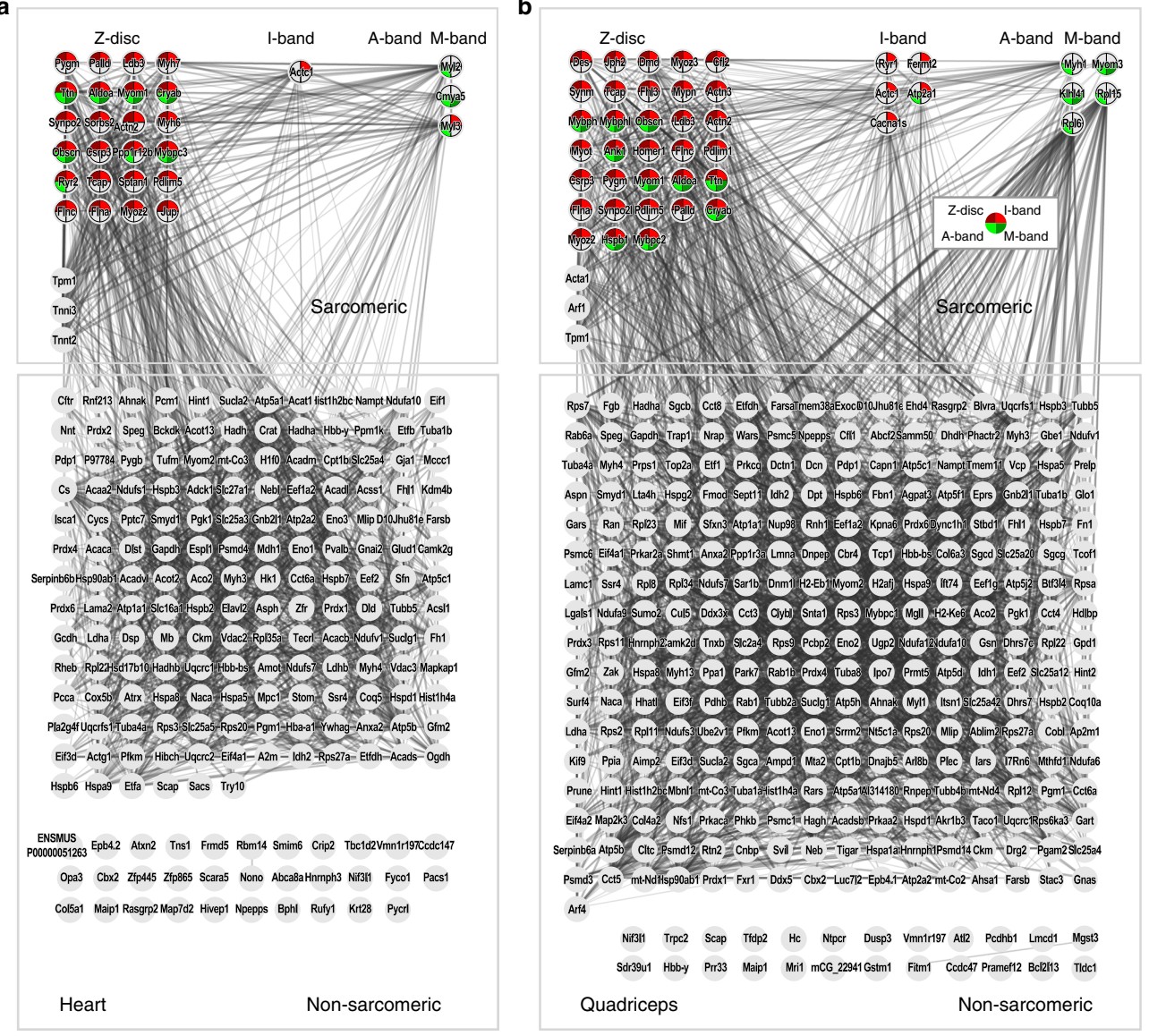

**Fig. 4 Sarcomeric and non-sarcomeric proteins associate with the myofilament backbone. a** Cardiac myofilament proteome with sarcomeric and non-sarcomeric proteins separated. **b** Quadriceps myofilament proteome. Gray lines represent known interactions derived from the String database, supporting the identified proteins as sarcomere associated. Colored segments represent the Gene Ontology terms Z-disc, I-band, A-band, and M-band. Proteins that do not interact are listed below the network.

Towards understanding the force–length relation of the sarcomere, we provide experimental support for the myosin/Z-disc sliding model[15] that helps explain the dependence of maximum contraction velocity on sarcomere length (Fig. 6c). The biotinylation could derive from integrated or soluble titin. The latter amounts to <20% of total titin and would have similar access to soluble actin and myosin. Although actin is more abundant with 27 lysines available for biotinylation (Actc1), we only find biotinylated myosin with 14 lysines in myosin light chain (Myl1) and >40 hits in 2 different sites. The amount of myosin biotinylation (second only to titin) together with the localization of myosin at the border to the Z-disc by immunofluorescence staining is consistent with biotinylation of myosin at the Z-disc in support of the myosin/Z-disc sliding model.

Our BioID approach provides a unique look at sarcomere physiology in vivo that cannot be achieved by conventional morphological analysis, where unloading and fixation could result in a hypercontracted sarcomere and artificial Z-disc penetration

of myosin. Here, positioning BioID at the edge of the Z-disc has allowed us to capture both myosin heavy and light chain proteins that pass the Z-disc boundary under physiological conditions in heart and skeletal muscle. A limitation of the in-vivo system is that not all control groups employed in cell culture are easily available (e.g., minus biotin). Here we used wild-type mice to distinguish endogenous ligase activity from titin-BioID activity. In cell culture, ligase alone is used to subtract proteins that are either nonspecifically pulled down and/or proteins with an affinity to the ligase itself. As more in vivo BioID models become available, it will be easier to control for these nonspecific binders.

Taken together, we provide an in-depth analysis of the sarcomeric proteome as it links to signaling, metabolism, and biomechanics, extend the worm-like chain model to describe titin's elastic properties and address the role of the Z-disc in the force–length relationship of the sarcomere, with implication for cardiac and skeletal muscle development, health, and disease. Adapting localization proteomics to the situation in vivo has the

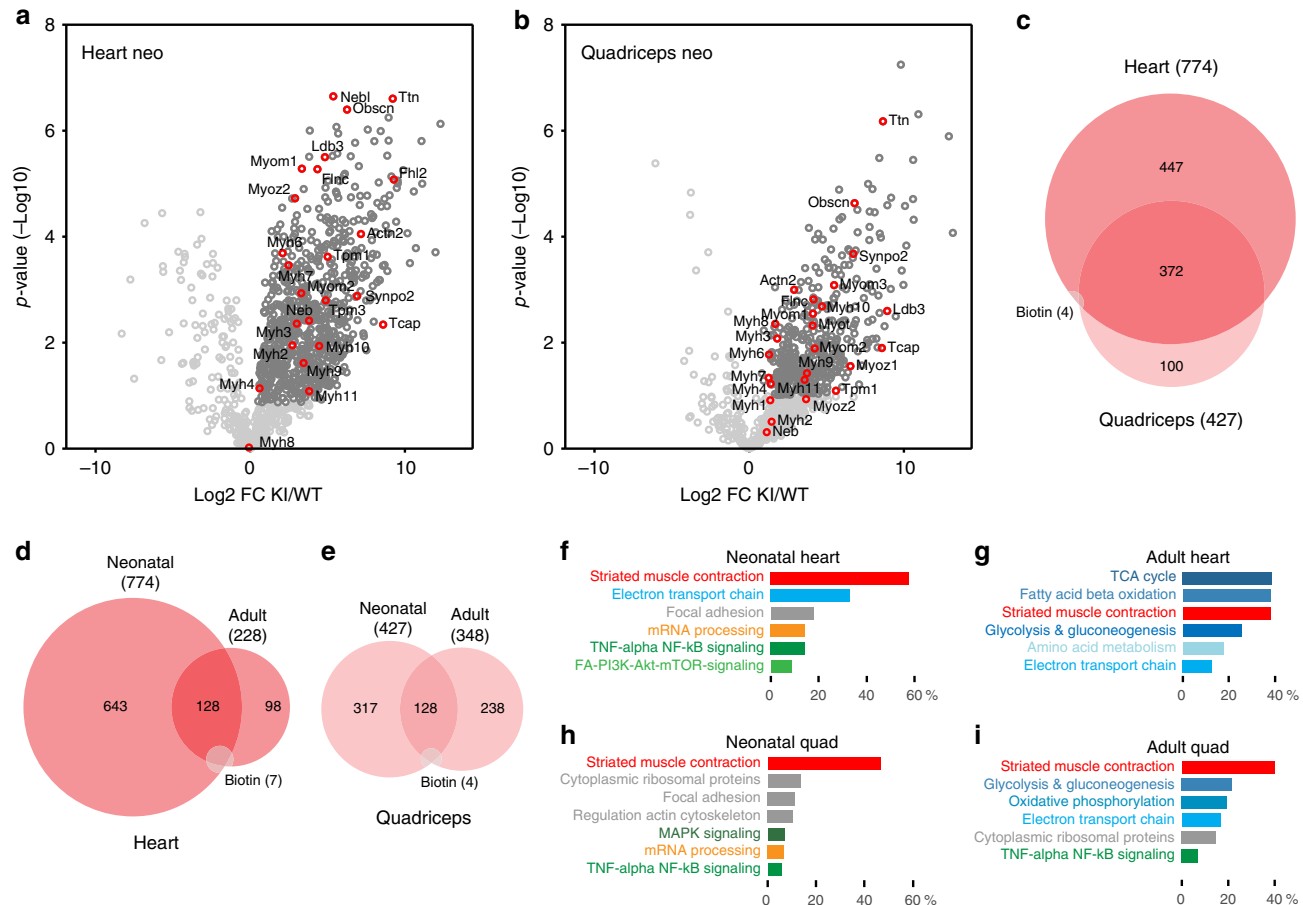

**Fig. 5 The sarcomeric neonatal vs. adult proteome.** Proteins that co-purify with biotinylated proteins after streptavidin pulldown of neonatal TiZ-BioID knock-in vs. WT heart (**a**) and quadriceps (**b**) link to partially overlapping signaling and metabolic pathways in the heart and skeletal muscle. Known sarcomeric proteins in red, significantly enriched in dark gray. **c** The neonatal sarcomeric proteome comprises >850 proteins, >350 shared between the heart and quadriceps. The transition from neonatal to adult involves a major switch in expression of sarcomere-associated proteins in the heart (**d**) and quadriceps (**e**). **f–i** Wikipathway analysis (% genes per term). The majority of sarcomere-associated proteins relates to striated muscle contraction (red) with a shift from structural (gray) and signaling pathways (green) to metabolism (blue) in the adult. In neonatal and adult quadriceps, ribosomal proteins are enriched (**h**, **i**), mTOR signaling is specific to the neonatal heart (**f**) vs. MAPK signaling in quadriceps. NF-κB signaling is enriched in all neonatal and quadriceps tissues (**f**, **h**, **g**). Source data are provided as a Source Data file.

potential to not only provide mechanistic insights into basic cellular functions, but also into processes that are primarily studied at the level of the organism such as embryonic development and ageing.

## Methods

**Statistics and reproducibility.** No statistical methods were used to predetermine sample size. The experiments were not randomized. The investigators were not blinded to allocation during experiments and outcome assessment. All experiments were performed at least twice unless otherwise noted. $P < 0.05$ was accepted as statistically significant. Data are displayed as mean ± SEM unless indicated otherwise. Data in Figs. 1c, 1d and 2i were replicated once with similar results. Data in Supplementary Fig. 1c were replicated >10× as part of standard genotyping procedures. Supplementary Figs. 1i–l and 2k–m were each replicated at least once with similar results.

**Animal models.** All experiments involving animals were performed following the rules for Animal Welfare of the German Society for Laboratory Animal Science and received ethical approval by the Landesamt für Gesundheit und Soziales (LAGeSo, Berlin). Animals were housed in individually ventilated cages with free access to food and water, constant 22 ± 2 °C temperature and 55 ± 10% humidity, and a 12 h : 12 h light/dark cycle (light from 6:30 a.m. to 18:30 p.m.).

The animal model to probe the Z-disc and titin interactome was generated using homologous recombination in ES cells and blastocyst injection[34] and backcrossed on a 129/S6 background (Taconic Biosciences). We inserted BirA downstream of Exons 28 at the transition of the Z-disc into the I-band—the identical position that we used for our integration of DsRed, as it did not interfere

with sarcomere assembly, remodeling, or mechanics[17]. Animals received continuous biotin supplementation (3.7 µg/ml added to the drinking water) to improve levels of free Biotin and facilitate protein biotinylation. Animals were sacrificed by cervical dislocation. Heart weight, body weight, and tibia length was measured from wild-type, heterozygous, and homozygous knock-in animals at the indicated timepoints in male and female animals. All experimental animals were male adults (12–15-week-old) or neonatals (P1/2). For controls, we used wild-type animals to distinguish unspecific endogenous biotinylation. As this is the first BioID knock-in mouse, there are no suitable controls available with biotinylation mediated by a fusion protein outside the sarcomere. Unlike in cell culture experiments, we are not able to control for biotin in our mice, as biotin deficiency is not tolerated in vivo[16].

**Genotyping.** Genomic DNA was prepared from ear tags following standard procedures. Proper integration of the BirA was verified by PCR using specific primers. The BirA construct into the titin locus was monitored by PCR using primers BirA-SeqFw (5′-CATCTCCAGAGGAATCGACAAG-3′) and rMA dsredKITV (5′-AAG CTTGATAAGGGATAGTCTTGGGCATAC-3′). Excision of the Neo selection and to distinguish between heterozygous and homozygous animals a PCR with the primers fDsRedrecF (5′-CAGCATCATGGTAAAGGCCATCAA -3′) and rDsRe-drecF (5′-CATTCAAATGTTGCCATGGTGTCC-3′) was done. The Flp recombinase allele was detected using primers Flp-for (5′-GTCACTGCAGTTTAAATA CAAGACG-3′) and Flp-rev (5′-GTTGCGCTAAAGAAGTATATGTGCC-3′).

**SDS-agarose electrophoresis and western blotting.** Tissues were collected, pulverized under liquid nitrogen using mortar and pestle, and extracted in titin sample buffer (8 M urea, 2 M thiourea, 3% SDS, 0.05 M Tris-HCl, 0.03% bromophenol blue, 75 mM dithiothreitol (DTT), pH 6.8) for 30 min on ice. Debris was

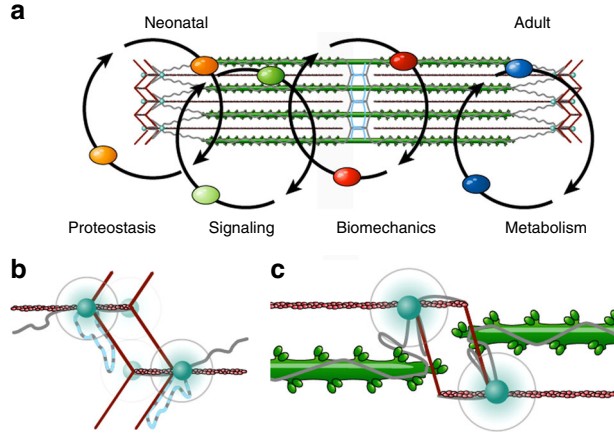

**Fig. 6 Remapping sarcomere structure and function. a** The sarcomere serves as a multi-protein hub that does not only determine the mechanical properties of striated muscle, but links to proteostasis and distinct signaling pathways in the neonatal heart and skeletal muscle vs. metabolic enzymes in adult tissue to meet local demand. **b** Revised model of actin/myosin/titin interactions at the Z-disc. The titin Z-disc loop (light blue) interacts with α-actinin (red rods). The biotin ligase (blue dot) is integrated between titin Ig7 and Ig8 folding back to the edge of the Z-disc. **c** As sarcomeres contract, myosin approaches and eventually penetrates the Z-disc at short physiological sarcomere lengths.

removed by centrifugation. Proteins were separated on agarose gels and blotted on polyvinylidene difluoride membranes. Primary BirA antibody were purchased from BioFront Technologies (Chicken Polyclonal Ab to *Escherichia coli* Biotin Ligase/BirA; BID-CP-100, 1 : 5000) and the Streptavidin–horseradish peroxidase (HRP) conjugate from GE Healthcare (RPN1231V, 1 : 1000). The secondary HRP-conjugated antibody (ECL Rabbit IgG, HRP-linked whole Ab (from donkey) Amersham NA934V, 1 : 5000) was detected by chemiluminescence staining with ECL (Supersignal West Femto Chemiluminescent Substrate; Pierce Chemical, Co.).

**Real-time PCR.** Heart and quadriceps tissue of all genotypes was collected, snap frozen, and grinded. Total RNA was isolated from tissue powder with Trizol followed by a cleaning step with the Qiagen RNeasy isolation Kit. To convert RNA into complementary DNA (cDNA), a reversed transcription PCR with a viral RNA-dependent DNA polymerase was performed using the RNA to cDNA Kit from Applied Biosystem or ThermoScientificTM RevertAid RT Kit according to the manufacturer's specifications. Taqman probe against the BirA was used to monitor the RNA expression in all genotypes (BirAfw 5′-ATCGGACT-GAGTCTGGTGAT-3′; BirArev 5′-GTACAGGTCATTAGGCCACTTC-3′; Bir-Aprobe FAM-5′-CTGAGAAAGCTGGGAGCCGACAAG-3′-TAMRA).

**Immunofluorescence staining.** Hearts and quadriceps from BirA homozygous and wild-type animals were dissected and prepared for immunofluorescence staining[35]. Flexor digitorum brevis muscle was isolated and transferred to MEM medium containing Biotin. One muscle was stretched with Minutien Pins on Sylgard dishes the other muscle was pinned without stretch as control. Stretch experiments were performed for 15 min at 37 °C. Contraction was induced by skinning the muscles first with 2% Triton X-100 for 30 min followed by incubation with 4 mM CaCl$_2$. The muscle was fixed with 4% paraformaldehyde (PFA) at room temperature, followed by fixation for 2 h at 4 °C. The muscles were transferred into 30% sucrose phosphate-buffered saline (PBS) solution overnight and embedded in TissueTek. Cryosections of 8 μm were fixed with 4% PFA at room temperature for 10 min, and permeabilized and blocked with 2% goat serum, 0.3% Triton X-100, and 2 % bovine serum albumin in PBS for 2 h. The incubation with the primary antibody (1 : 50 anti-α-actinin, Sigma-Aldrich EA-53; 1 : 50 anti-Myh8, Invitrogen PA5-72846; 1 : 50 anti-Myosin, Sigma-Aldrich M4276; 1 : 50 anti-Nebl, SCBT sc-393784; 1 : 50 anti-Pgam2, Abcam ab97800; 1 : 50 anti-titin BirA, BioFront Technologies) or Streptavidin Star conjugate (1 : 200, Abberior) was performed at 4 °C overnight followed by labeling with a fluorescent secondary antibody (1 : 200 goat anti-chicken, goat anti-mouse, or goat anti-rabbit STAR from Abberior) for 2 h at room temperature or overnight at 4 °C.

Images were acquired with a confocal laser-scanning microscope (LSM710, Carl Zeiss) with a Plan-Apochromat ×63/1.4 oil Ph3 objective. Super-resolution imaging was performed with a 3D-STED microscope (Abberior Instruments) and a ×100 oil objective (UPLANSAPO) or a TCS SP8 STED microscope (Leica) with a HC PL APO C2S ×100 oil objective (1.4 NA) used in 2D mode[17]. For imaging with the Leica STED system, Abberior STAR 635P was excited at a wavelength of 635 nm

and fluorescence was detected between 650 nm and 700 nm. Abberior STAR 580 was excited at 580 nm and fluorescence was detected between 590 nm and 630 nm. For the Abberior system, the fluorescence excited at 635 nm was detected between 605 and 625 nm, and the fluorescence excited at 580 nm between 650 and 720 nm. For both dyes and systems, STED imaging was performed with a 775 nm depletion laser. Imaging with the Leica STED system was executed with a gating between 0.5 ns and 6 ns, and images were acquired with a pixel size of 23 nm × 23 nm and a scanning speed of 600 Hz (pixel dwell time 0.4 μs). With the Abberior system, the depletion laser was used with a laser power of 15% and images were made with a pixel size of 15 nm × 15 nm. All images not labeled as STED are confocal images. Abberior STAR 635 P signal is displayed in red, Abberior STAR 580 in green. Images were analyzed using ImageJ Fiji Software. Traces are displayed as mean (opaque) ± SEM (SEM, transparent lines).

**Pulldown experiments.** For the identification of protein complexes, the protocol from Roux et al.[16] was modified. Fifty milligrams of cryo-fractured tissue powder was resuspended with 0.5 ml BioID lysis buffer (50 mM Tris pH 7.5; 500 mM NaCl; 0.4% SDS; 5 mM EDTA; 1 mM DTT; 2% Triton X and 1× PICS I). The samples were incubated for 60 min on ice and snapped in between followed by sonication. To remove insoluble material the samples were centrifuged for 10 min at 12,000 r.p.m. at 8 °C. The supernatant was transferred into a new tube and incubated with the beads (MyoOne Streptavidin C1; Invitrogen) overnight[16]. Beads were collected with a magnetic rack and washed 2 times for 10 min with 0.1% deoxycholate; 1% NP-40; 500 mM NaCl; 1 mM EDTA and 50 mM HEPES pH 7.5 at 4 °C, 2 times washed with 250 mM LiCl, 0.5% NP-40, 0.5% deoxycholate, 1 mM EDTA, and 10 mM Tris pH 8.1 at 4 °C, and 2 times washed with 50 mM Tris pH 7.4 and 50 mM NaCl at 4 °C. Beads were resuspended in 20 μl urea buffer (6 M urea, 2 M thiourea, 10 mM HEPES pH 8.0), reduced in 10 mM DTT solution, followed by alkylation using 40 mM chloroacetamide. Samples were first digested with 1 μg endopeptidase LysC (Wako, Osaka, Japan) for 4 h and, after adding 80 μl 50 mM ammonium bicarbonate (pH 8.5), digested with 1 μg sequence-grade trypsin (Promega) overnight. The supernatant was collected and combined with the supernatant from an additional bead wash in 50 mM ammonium bicarbonate (pH 8.5). Samples were acidified with formic acid and peptides were desalted using StageTip purification[36].

To capture biotinylated peptides we use a modified anti-Biotin AB protocol[19]. Complete heart and quadriceps cryo-fractured tissue powder was lysed for 1 hour at 4 °C (8 M urea, 50 mM Tris-HCl pH 8.0, 150 mM NaCl, 1 mM EDTA, Protease inhibitor PICS I, 10 mM sodium azide, 10 mM sodium ascorbate). The samples were sonicated samples 3 and the protein concentration was determined using the bicinchoninic acid (BCA) assay. To reduce the proteins 5 mM DTT was added for 45 min at room temperature (RT) followed by carbamidomethylation with 10 mM iodoacetamide for 30 min at RT in the dark. To reduce the urea concentration to 2 M we used 50 mM Tris-HCl pH 8. The samples were digested overnight at 25 °C with sequencing-grade trypsin at an enzyme:substrate ratio of 1 : 50 or 1 : 100. To remove the insoluble material the samples were centrifuged at 4000 × g for 5 min at 4 °C until the supernatant was clear. The peptide concentration was measured with BCA assay and the samples were acidified with formic acid (1% final concentration) and desalted on a 500 mg C18 Sep-Pak SPE cartridge (Waters). Before the cartridges were washed 1× with 5 ml of 100% MeCN, 1× with 5 ml 50% MeCN:0.1% FA, 4× with 5 ml of 0.1% TFA. After sample loading, the cartridges were washed 3× with 5 ml of 0.1% TFA and then 1× with 5 ml of 1% FA. For elution of the peptides we used 6 ml of 50% MeCN:0.1% FA and dried by vacuum centrifugation (possible to store the peptides at −80 °C). For the enrichment of biotinylated peptides, a biotin antibody was used (anti-biotin ImmuneChem Pharmaceuticals, Inc.; ICP0615; 50 μg per sample). The antibody bound agarose beads were washed 3× in IAP buffer (50 mM MOPS pH 7.2, 10 mM sodium phosphate, and 50 mM NaCl). After washing, biotinylated peptides were eluted with 50 μl of 0.15% TFA, cleaned up, and stored on StageTips[36].

**LC-MS/MS analyses.** Peptides were eluted using Buffer B (80% Acetonitrile and 0.1% formic acid) and organic solvent was evaporated using a speedvac (Eppendorf). Peptide samples were diluted in Buffer A (3% acetonitrile and 0.1% formic acid) and separated on a 20 cm reversed-phase column (ReproSil-Pur C18-AQ; Dr. Maisch GmbH) using a 98 min or 200 min gradient with a 250 nl/min flow rate of increasing Buffer B concentration (from 2% to 60%) on a high-performance liquid chromatography system (ThermoScientific). Peptides were measured on a Thermo Orbitrap Fusion or Q Exactive Plus instrument (Thermo). On the Orbitrap Fusion instrument, peptide precursor survey scans were performed at 120 K resolution with a $2 \times 10^5$ ion count target. Tandem MS was performed by isolation at 1.6 m/z with the quadrupole, higher energy collisional dissociation fragmentation with normalized collision energy of 32, and rapid scan MS analysis in the ion trap. The MS2 ion count target was set to $1 \times 10^4$ and the max injection time was 300 ms. The instrument was run in top speed mode with 3 s cycles. The Q Exactive Plus instrument (Thermo Fisher Scientific) was operated in the data-dependent mode with a full scan in the Orbitrap (70 K resolution; $3 \times 10^6$ ion count target; maximum injection time 50 ms) followed by top 10 MS2 scans using higher-energy collision dissociation (17.5 K resolution, $1 \times 10^5$ ion count target; 1.6 m/z isolation window; maximum injection time: 50 ms for streptavidin enriched samples and 250 ms for anti-biotin antibody enriched samples).

**Data analyses**. Raw data were processed using MaxQuant software package (v1.5.1.2)[37]. The internal Andromeda search engine was used to search MS2 spectra against a decoy mouse UniProt database (MOUSE.2017-01) containing forward and reverse sequences, including the sequence for the Titin-BirA construct. The search included variable modifications of methionine oxidation and N-terminal acetylation, deamidation (N and Q), biotin (K) and fixed modification of carbamidomethyl cysteine. Minimal peptide length was set to seven amino acids and a maximum of two missed cleavages was allowed. The false discovery rate (FDR) was set to 1% for peptide and protein, and site identifications. To filter for confidently identified peptides, the MaxQuant score was set to a minimum of 40. Unique and razor peptides were considered for quantification. Retention times were recalibrated based on the built-in nonlinear time-rescaling algorithm. MS2 identifications were transferred between runs with the "Match between runs" function of MaxQuant (only within replicates of the same batch and tissue). IBAQ intensities were calculated using the built-in algorithm. The resulting proteinGroups and biotinSites text files were filtered to exclude reverse database hits, potential contaminants, and proteins only identified by site. Statistical data analysis was performed using Perseus software (v1.6.2.1)[38].

For the streptavidin protein pulldown data biological replicates for each genotype were defined as groups and proteinsGroups identifications with <3 valid values (>0) for at least one group were filtered out. After log2 transformation, missing values were imputed with random values taken from a normal distribution with 0.3× the SD of the measured, logarithmized values, down-shifted by 1.8 SDs. Differences in protein abundance between BirA-titin samples and wild-type control samples were calculated using two-sample Student's $t$-test. Proteins enriched in the BirA-titin group and passing the significance cut-off (permutation based FDR < 5%, minimum three peptides identified) were considered titin-associated proteins. GO term enrichment analyses for titin-associated proteins were perfomed using Cytoscape and the ClueGO plugin[39,40].

BirA-Titin-specific biotinylated peptides and their corresponding sites were extracted from the biotinSite text file, using a minimum of three valid values (Intensity > 0) in the BirA-Titin group and at least two more valid values than in the corresponding wild-type group. Annotated MS2 spectra for confident biotinylated peptides were extracted using MaxQuant Viewer application.

**Reporting summary**. Further information on research design is available in the Nature Research Reporting Summary linked to this article.

## Data availability
The data that support the findings of this study are available from the article and Supplementary Information files, or from the corresponding author upon request. Proteomics data have been deposited to PRIDE server under accession code PXD017341 [http://proteomecentral.proteomexchange.org/cgi/GetDataset?ID=PXD017341]. Uniprot database MOUSE.2017-01, STRING Database Version 10.5, and the Gene Ontology Database retrieved from https://doi.org/10.5281/zenodo.2529950 have been used in this study. Source data are provided with this paper.

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

## Acknowledgements

This work was funded by grants from Deutsche Forschungsgemeinschaft (DFG, German Research Foundation) FOR1352, Go865, and CRC 1002-S02, the European Research Council grant StG282078, the "Bundesministerium für Bildung und Forschung" grant CaRNAtion, and the German Center for Cardiovascular Research (DZHK), Berlin (to M.G. and to S.E.L.). We thank Janine Fröhlich and Carmen Judis for expert technical assistance, the transgenic core facility for the ES-cell injection, the imaging facility for access to their microscopes, and support with data acquisition (all MDC).

## Author contributions

F.R., C.F., M.K., J.H., M.H.R., T.K., and E.W. performed the experiments. F.R. and C.F. generated the animal model with support from the MDC core facility, designed the animal experiments, performed the biotin pulldowns, managed the project, and visualized the proteomics data. F.R., E.W., and J.H. performed the morphological analysis. M.K. generated, analyzed, and visualized proteomics data. J.L.C. supported the modeling of the Z-disc structure. S.E.L. and P.M. contributed to experimental design and supervision of the project. M.G. conceived the project, analyzed data, generated display items, and wrote the manuscript with input from all authors.

## Competing interests

The authors declare no competing interests.
