## [Peer Review File · Nature Communications]

Reviewers' comments:

Reviewer #1 (Remarks to the Author):

This manuscript by Rudolph et. al. describes the generation and use of a mouse model in which BioID has been knocked into the titin gene to identify protein-protein associations of this large structural protein that functions in muscle sarcomeres. These mice were used for BioID pulldowns using two different methods, one that identified sites of protein biotinylation to map proximities of protein domains with high resolution and another that pulled down all biotinylated proteins to map protein proximities. These pulldowns were performed from both heart and quadricep muscle and compared between newborn and adult tissues. Overall the generation of the mice appears well conceived and the experimental design was well thought out. There appears to be some confusion by the authors as to the interpretation of the conventional BioID pulldowns as somehow being able to isolate intact and stable protein complexes, in comparison to the method to capture biotinylated peptides which yielded far fewer detected proteins. Overall, this is a basic proof of principle application of BioID in a mouse where the ligase has been fused to an endogenously expressed protein. The identified proteins associated with titin should be of value to investigators in that field.

The duration of biotin administration should be mentioned in the methods.

Consider citing PMID 29249144 along with the Carr paper for the anti-biotin method since they were effectively independently co-developed with one coming out just before the other.

Could the authors show a full Western blot probed for biotinylated protein to show the extent of biotinylation and distribution of those proteins by MW.

The authors say: 'To extend our approach of in vivo colocalization proteomics from Z-disc titin to the sarcomere, we used an adapted streptavidin based pull-down protocol eliminating the initial trypsin digest to enrich entire sarcomere complexes.' What it looks like they are describing is in fact a normal BioID pulldown experiment that captures proteins biotinylated by BioID-titin. With the buffers used in the lysis and washing there should be no remaining protein-protein interactions, or complexes (500mM salts, substantial detergents, etc.). These conditions are designed to denature proteins allowing just the biotinylated ones to be selectively isolated. My suspicion is that the anti-biotin pulldown of tryptic peptides that only identified 14 proteins was simply very inefficient and is only detecting the most abundant biotinylated peptides. It doesn't change the results for those anti-biotin AB experiments, in that it can inform on the relative proximities of these proteins to the ligase, but it doesn't mean that those were the only proteins biotinylated. Both the Carr and the

Pandey (PMID 29249144) papers showed that there was relatively similar detection of proteins with the conventional pulldown and the anti-biotin method with considerable overlap in those proteins detected. The discrepancy here suggests a technical issue with the capture of biotinylated peptides.

The authors should acknowledge that all BioID experiments should have a control, typically ligase-alone, or something similar to subtract proteins that are either non-specifically pulled down and/or proteins with an affinity to the ligase itself, of which there are many. Given the cost and effort to generate these mice, the lack of such a control can be forgiven; however, this issue must be addressed to qualify the limitations of the results.

The authors need to provide the MS result in a excel file with the identities of the proteins and relative abundance as is conventional for these types of studies.

In the methods there is mention of 'Global proteome of neonatal and adult mouse tissue by LC-MS'. What is this is reference to?

Reviewer #2 (Remarks to the Author):

The manuscript titled "Deconstructing sarcomeric structure/function relations in titin-BioID knock-in mice" by Rudolph et al. provides an excellent resource for the interactome or signalosome of striated muscle sarcomere. And it has the potential to enable further studies with BioID in the muscle field following this proof-of-concept work. The authors confirm that the knock-in mice are in general healthy and therefore a suitable model to dissect even transient protein-protein interactions at the Z-disc in the living animal. This elegant approach harbours plenty of information for functional follow-up studies and thereby is interesting for a wider audience in the muscle field and – regarding the method – also beyond.

The authors aim to connect the comprehensive biochemical work on heart and quadriceps in the adult and neonatal mouse to achieve a better understanding of structure-function relationship of titin with other proteins at striated muscle sarcomere. The biochemical data is an excellent source providing information about (potential) interactions on the protein level – which should dominantly happen at the z-disc (although the free titin pool can also contribute). These interactions can be either transient or between well-known and stable interaction partners. Especially, the potential to identify novel (transient) interaction partners could be great value also for future studies in the muscle field and other fields. On the other hand, the authors need to pay more attention to present

the data completely and convincingly, incl. data analysis and data presentation, in particular while establishing the relative positioning of various sarcomeric proteins investigated here. The comments in detail as following:

The authors try to support their biochemical findings by immunofluorescence microscopy employing super-resolution STED microscopy. To achieve a convincing and balanced experimental support of their favoured explanations outlined in the manuscript, they should present especially their STED imaging data more detailed and include additional controls. They should also consider adding additional experiments supporting their favourite models of sarcomere interaction and muscle function.

In detail:

The authors should provide more information about the STED imaging in the materials and methods section:

- They state twice (page 19 and 20) that the method “has been described” elsewhere but do not provide any reference. The references should be added and the authors might want to consider to briefly outline the experimental procedure to avoid confusion, e.g. about the thickness of the tissue sections/slices used, STED mode used (2D or 3D – see also below) etc.

- In the materials and methods, the author mentioned that “imaging was performed with a 3D-STED microscope” However, the pixel sizes used for image acquisition would be optimal for so-called 2D-STED. Based on this, I assume that 2D-STED has been used and for my interpretation of the results.

General comment to fluorescence representation in all figures:

- The size of the images is rather small and capturing a large sample area, making it hard if not impossible for the reader to confirm the described localization pattern in the text. The authors have reserved a lot of space for depicting graphs of proteomic data, which could also be presented in a list or a table, while the key imaging-experiments for validating the screen are squeezed into very limited space and provide no real chance for the reader to see the confirmation but only to believe it. The images should be enlarged and panels displaying multiple region of interested be shown at larger magnification (zoom-in). For example, In Fig. 1f or h (see also image attached). Here, after zooming in on a screen one can really see that there could be indeed three lines of titin and the traces become more believable. In addition, for each image, individual channel images and then a

merge should be presented. Panels where half the image is just one channel and the other half is a merge can be very misleading (see Fig. 1 f and h).

- For each multi-colour image, at least one, but better multiple line profiles should be displayed for all fluorescent markers.

- The authors should indicate in the materials and methods section, which fluorophores correspond to which colour code in the figures (e.g. if Star 580 is always encoded in green).

- Stating partial co-localization can be a dangerous route. Therefore, it is very important to provide the reader with all details and control experiments. The authors should also consider that the interaction could be transient or the protein be biotinylated by the non-integrated titin pool and thereby no co-localization would be detectable in immunofluorescence. The current presentation of the imaging data does neither confirm nor exclude partial-co-localization of Myosin / MyH8, Pgam2 and Neb1 (see also per figure comments below).

A more extensive (semi-) quantification of the signals (e.g. by averaging multiple line profile on different samples and displaying SD or SEM) would also improve the manuscript.

Detailed comments per figure:

- Fig. 1 f-i: Line Profiles should be shown for both colours (see also above). Both channels should be first displayed separately and then with the merge.

- Fig. 2 h/j: The authors should adhere to the general practice to draw line profiles in a consistent manner. This is usually perpendicular to the structure of interest, here the biotinylation region.

- Ext. Data Fig. 2: k-o partial co-localization is hard to confirm by the reader due to the small size of the images and lack of line profiles (see general comments above). Furthermore, in k, the Nebulette signal at the sarcolemma is saturated and preventing to estimate the fraction of protein at the z-disk. The authors might want to clarify this by showing non-normalized line profiles. Signal crosstalk between the STED dyes used in this study has been observed on other occasions when the signal intensity was substantially stronger in one channel than the other. The authors should briefly clarify the used fluorophore combination and add the respective controls to the figure (e.g. by presenting representative single staining of the structures imaged with the same microscope settings).

In my opinion, the authors have undertaken a great task of localising sarcomeric proteins with advanced light microscopy, and this can be really helpful for the field. To establish precisely the co-localization pattern and degree of overlap (and make this easily visible to all readers), images displaying the biotinylation band and an immunostaining for a protein that resides

a) Exclusively outside the z-disc

b) Exclusively at the z-disk and

c) are present (to varying degree) in both regions

should be added.

A relative “sarcomeric map” should be made (see also Kelu et al., Dev Biol, 2017, <https://doi.org/10.1016/j.ydbio.2017.03.031>; Figure 9) to support the reader with the relative position of any protein that was measured.

Although, this work has the full potential to become pioneering work. The currently included experiments and especially the data presentation provokes the feeling of being partially unfinished and causes unnecessary ambiguity. In my opinion, additional experiments could strengthen, the overall message of this work and I would like to urge the authors to consider including the following experimental methodologies:

1) To support the biochemical data further, a super-resolution immune-fluorescence mapping of titin structure by employing antibody directed against various Ig domains of titin (that can possibly be resolved by STED) would be more than supportive for the findings presented. This could establish a better overview of the confirmation of titin in stretched and non-stretched muscle and provide additional evidence for the biotinylation pattern of titin.

2) A mapping of myosin in respect to the biotinylation band using different antibodies recognizing the rod, the globular domain and the light chains in non-stretched and in stretched muscle should provide a detailed view to what extent the myosin is indeed coming close or penetrating the z-disc. I can imagine that antibodies recognizing the light chain or the globular domain could provide a better and more convincing overlap with the biotinylation band. In addition, this would remove ambiguity that the biotinylation band found ~200nm away from the z-disc in the stretched muscle is formed by other proteins than myosin.

3) Single muscle fibres have been shown to provide high quality STED images. The authors might want to extend their study using these kind of samples.

(<https://www.ahajournals.org/doi/pdf/10.1161/CIRCRESAHA.112.274530>)

Minor comments but important aspects of the manuscript that can be changed for the ease of understanding of a broader audience or non-titin specialist skeletal muscle biologists:

- 1) Throughout the manuscript, the supplementary figures should be named as “supplementary” not as “extend figures” or can be presented the other way around but consistently
- 2) Page 5 it should be traces 1g vs i NOT 1e vs i
- 3) Please mention if an image is confocal, see Fig 1c
- 4) Page 6, the authors reported to have found “14” biotinylated proteins in the main text and “15” in the extended Data Fig.2 legend.
- 5) Supplementary Table 2: Neonatal Quad vs Neonatal Heart, there is a significant number of collagen proteins missing from Neonatal Quad, even though collagens have been shown to be essential for skeletal muscle development. This should be discussed and what could this mean in terms of the developmental differences between the two muscle types?
- 6) Page 10: Second paragraph “As we integrate colocalization information over time”: it is not necessarily clear if author have done any time lapse microscopy or want to refer to their biotinylation approach.

Marko Lampe with substantial support by

Muzamil Majid Khan

Reviewer #3 (Remarks to the Author):

The authors present a very interesting approach for testing protein-protein interactions in the cardiac and skeletal muscle sarcomere. By generating a BioID mouse with a biotin-ligase located in the Z-disk region of titin the authors are now able to monitor titin interactions with highly improved spatio-temporal resolution, compared to previous approaches.

The experiments are well-performed, statistical analyses are appropriate, description of the methods is short but precise.

Key findings of the paper are:

- identification of 478 sarcomeric or sarcomere associated proteins, which is much more than suggested before.
- identification of titin Ig8/9 as putative hairpin forming structures that may influence titin-based elastic properties of the sarcomere

- confirmation of previous hypotheses suggesting Z-disk penetration by myosin filaments during contraction/relaxation cycle

- Description of titin as a dynamic signaling hub for signal transduction and metabolism, depending on developmental state.

The manuscript is packed with excellent original data and provides extensive supplementary material. Some of the data are confirmatory, others really novel and exciting.

Description and interpretation of these large data sets in a comprehensive manner and for a large readership is indeed challenging. The reader is overwhelmed by the different data that range from general protein interaction studies, structural titin-spring studies to developmental changes in metabolism. Not to mention the data from proteome analyses.

At some points the selection of data shown in the graphs seems random, e.g. Figure 2. Here, biotinylated metabolic enzymes are presented, including Ldb3, Phtf1 and Pgam2, but most of the proteins shown are not really discussed later in the manuscript.

For the reader it is very hard to follow all these different lines of evidences without getting lost in details of the figures. One suggestion could therefore be to focus the manuscript body to the main findings and move some of the detailed graphs to the supplement or to an independent manuscript.

Maybe then it would be possible to go into more detail when explaining e.g. the titin hairpin theory. I think this is really exciting!!

I was a bit lost when looking for the additional subset of identified SR-titin interactions, mentioned on page 7. This could also be described in more detail.

I also feel that a few sentences on the principle of integration of a BioID should be added to the introductory section.

However, the discussion is really well structured and very well written and thus helps understanding the different findings of the manuscript.

Martina Krüger

Dear members of the editorial board, dear reviewers,

Thank you for the evaluation of our work and the opportunity to improve on our manuscript with additional experimental data (adding two more authors) and to better explain our approach.

Despite complications with the Max-Delbrück-Center shutdown since 3 weeks (including the imaging core facility and our animal colony, where all breedings had to be discontinued), we were able to obtain additional experimental data, and hope that the additional information and data included in the revised manuscript appropriately addresses the concerns of the reviewers.

As this is the first in vivo application of localization proteomics with physiological expression, we are concerned about the competition and thus greatly appreciate your timely revision.

Best regards,

Michael Gotthardt - for the authors

Reviewer #1 (Remarks to the Author):

Comment Reviewer 1-1:

This manuscript by Rudolph et. al. describes the generation and use of a mouse model in which BiOD has been knocked into the titin gene to identify protein-protein associations of this large structural protein that functions in muscle sarcomeres. These mice were used for BiOD pulldowns using two different methods, one that identified sites of protein biotinylation to map proximities of protein domains with high resolution and another that pulled down all biotinylated proteins to map protein proximities. These pulldowns were performed from both heart and quadricep muscle and compared between newborn and adult tissues. Overall the generation of the mice appears well conceived and the experimental design was well thought out. There appears to be some confusion by the authors as to the interpretation of the conventional BiOD pulldowns as somehow being able to isolate intact and stable protein complexes, in comparison to the method to capture biotinylated peptides which yielded far fewer detected proteins. Overall, this is a basic proof of principle application of BiOD in a mouse where the ligase has been fused to an endogenously expressed protein. The identified proteins associated with titin should be of value to investigators in that field.

Response:

Thank you for the positive evaluation of our work. Towards the interpretation of our pulldown experiments – we greatly appreciate the helpful comments on the possibility to isolate protein complexes, which made us realize that we did not sufficiently explain the major implications that working *in vivo* at physiological expression levels has. In brief, we worked to increase stringency of our pulldown and despite high salt and 1mM DTT, we work with high protein concentration in a tissue under evolutionary selection for strong protein-protein interactions to fortify the sarcomere lattice and physiological oxidative stress that leads to the formation of disulfide bonds. The resulting differences between striated muscle tissue and cell culture system contribute to the interpretation of our *in vivo* data after peptide vs. protein pulldown, which we now explain better in the revised manuscript (details in response to comment 1-5).

Comment Reviewer 1-2:

The duration of biotin administration should be mentioned in the methods.

Response:

We are sorry we did not explain this well – these are challenging experiments so we take the approach to generate as much signal as we can and thus provide the highest amount of label possible, which we achieve through continuous feeding with biotin: The animals get their biotin supplement from before

they are born. We feed the mothers and keep feeding the offspring and feed the adults. They don't know any food but the biotin fortified chow.

We changed:

Animals received biotin supplementation (3.7 µg/ml added to the drinking water) to improve levels of free Biotin and facilitate protein biotinylation.

To

Animals received continuous biotin supplementation throughout their lifetime (3.7 µg/ml added to the drinking water) to improve levels of free Biotin and facilitate protein biotinylation.

Page 19 line 17

Comment Reviewer 1-3:

Consider citing PMID 29249144 along with the Carr paper for the anti-biotin method since they were effectively independently co-developed with one coming out just before the other.

Response:

We now cite the Carr lab (Udeshi et al. Nature Methods 2017) and the Pandey lab (Kim et al. J Proteome Res 2018).

Page 4 line 14.

Comment Reviewer 1-4:

Could the authors show a full Western blot probed for biotinylated protein to show the extent of biotinylation and distribution of those proteins by MW.

Response:

The full western blot is now included in the extended data figure 1j. All complete gels are now available in the source data file. (Rudolph ncomms-source-data.xlsx)

Extended data Page 1 Figure 1j

Comment Reviewer 1-5:

The authors say: 'To extend our approach of *in vivo* colocalization proteomics from Z-disc titin to the sarcomere, we used an adapted streptavidin based pull-down protocol eliminating the initial trypsin digest to enrich entire sarcomere complexes.' What it looks like they are describing is in fact a normal BioID pulldown experiment that captures proteins biotinylated by BioID-titin. With the buffers used in the lysis and washing there should be no remaining protein-protein interactions, or complexes (500mM salts, substantial detergents, etc.). These conditions are designed to denature proteins allowing just the biotinylated ones to be selectively isolated. My suspicion is that the anti-biotin pulldown of tryptic peptides that only identified 14 proteins was simply very inefficient and is only detecting the most abundant biotinylated peptides. It doesn't change the results for those anti-biotin AB experiments, in that it can inform on the relative proximities of these proteins to the ligase, but it doesn't mean that those were the only proteins biotinylated. Both the Carr and the Pandey (PMID 29249144) papers showed that there was relatively similar detection of proteins with the conventional pulldown and the anti-biotin method with considerable overlap in those proteins detected. The discrepancy here suggests a technical issue with the capture of biotinylated peptides.

Response:

We agree that there are obvious technical difficulties with the application of BioID to the physiological situation *in vivo*. This is reflected in the lack of prior art now more than 8 years after the original description of the BioID approach. One technical challenge is the preparation of tissue lysates suitable for BioID. Unlike cell-lines in culture, which have a lower protein content and are easily accessible to detergent, cardiomyocytes *in vivo* are densely packed with tightly interacting sarcomeric proteins under constant mechanical and metabolic stress.

Our results suggest that data obtained from cardiomyocytes *in vivo* are different from data obtained so far in tissue culture (including our own published and unpublished work). This relates to the amount of label from physiologically expressed fusion proteins and to the dense structure of the filament system, but also to the accumulation of disulfide bridges in non-dividing cells under constant oxidative and mechanical stress. For Biotin-Antibody Capture as compared to Streptavidin we used very stringent reducing conditions with increased temperature and 5mM DDT to remove any residual disulfite bridges between proteins. To adapt the protocol to the tissue, we used cryofractured tissue powder and to optimize the protocol for stringency we opted for high salt and detergent concentrations.

Page 4 line 10-16.

Comment Reviewer 1-6:

The authors should acknowledge that all BioID experiments should have a control, typically ligase-alone, or something similar to subtract proteins that are either non-specifically pulled down and/or proteins with an affinity to the ligase itself, of which there are many. Given the cost and effort to generate these

mice, the lack of such a control can be forgiven; however, this issue must be addressed to qualify the limitations of the results.

Response:

This is an important comment and we have extended the methods section to explain experimental restrictions associated with moving BioID *in vivo*. Controls are crucial, but not all controls available in the tissue culture system are available *in vivo* (e.g. there is no possibility for a non-biotin control as the animals would die). We have used wildtype animals as controls, as these allow us to distinguish endogenous biotin ligase activity from biotin ligase added to titin. In addition, we revisited published BioID experiments in myoblasts (C2C12 cells). Here, Schmidtman et al. PMID: 27676121 did not find any sarcomeric proteins that overlapped with our data.

We now prominently discuss the issue of controls at the final paragraph of the discussion just before the summary.

Methods: Page 19 line 23- Page 20 line 2.

Discussion: Page 12 line 4 to 10.

Comment Reviewer 1-7:

The authors need to provide the MS result in a excel file with the identities of the proteins and relative abundance as is conventional for these types of studies.

Response:

All this is now available in the source data file (Rudolph ncomms-source-data.xlsx). In addition, all proteomics data have been uploaded to the PRIDE server under accession code PXD017341.

Reviewer account - Username: reviewer67313@ebi.ac.uk

Password: 4VAqaTSf

Comment Reviewer 1-8:

In the methods there is mention of 'Global proteome of neonatal and adult mouse tissue by LC-MS'. What is this is reference to?

Response:

Thank you for noticing – this was left over from a different manuscript used as a reference. We have removed the section.

Reviewer #2 (Remarks to the Author):

Comment Reviewer 2-1:

The manuscript titled “Deconstructing sarcomeric structure/function relations in titin-BioID knock-in mice” by Rudolph et al. provides an excellent resource for the interactome or signalosome of striated muscle sarcomere. And it has the potential to enable further studies with BioID in the muscle field following this proof-of-concept work. The authors confirm that the knock-in mice are in general healthy and therefore a suitable model to dissect even transient protein-protein interactions at the Z-disc in the living animal. This elegant approach harbours plenty of information for functional follow-up studies and thereby is interesting for a wider audience in the muscle field and – regarding the method – also beyond.

The authors aim to connect the comprehensive biochemical work on heart and quadriceps in the adult and neonatal mouse to achieve a better understanding of structure-function relationship of titin with other proteins at striated muscle sarcomere. The biochemical data is an excellent source providing information about (potential) interactions on the protein level – which should dominantly happen at the z-disc (although the free titin pool can also contribute). These interactions can be either transient or between well-known and stable interaction partners. Especially, the potential to identify novel (transient) interaction partners could be great value also for future studies in the muscle field and other fields.

Response:

Thank you for your encouraging comments, highlighting the potential impact on a wider audience.

Comment Reviewer 2-2:

On the other hand, the authors need to pay more attention to present the data completely and convincingly, incl. data analysis and data presentation, in particular while establishing the relative positioning of various sarcomeric proteins investigated here. The comments in detail as following:

The authors try to support their biochemical findings by immunofluorescence microscopy employing super-resolution STED microscopy. To achieve a convincing and balanced experimental support of their favoured explanations outlined in the manuscript, they should present especially their STED imaging data more detailed and include additional controls. They should also consider adding additional experiments supporting their favourite models of sarcomere interaction and muscle function.

Response:

This point is well taken and we tried to accommodate all requests as outlined below. Although we went beyond the state of the art in the localization proteomics literature to validate our data by immunostaining, we agree that there was certainly additional room for improvement of the presentation. We tried to accommodate both the suggestions from reviewer 2 for additional and larger

display items and the comments from reviewer 3 for better structure with our complex and diverse datasets.

Comment Reviewer 2-3:

In detail:

The authors should provide more information about the STED imaging in the materials and methods section:

- They state twice (page 19 and 20) that the method “has been described” elsewhere but do not provide any reference. The references should be added and the authors might want to consider to briefly outline the experimental procedure to avoid confusion, e.g. about the thickness of the tissue sections/slices used, STED mode used (**2D** or 3D – see also below) etc.
- In the materials and methods, the author mentioned that “imaging was performed with a 3D-STED microscope” However, the pixel sizes used for image acquisition would be optimal for so-called 2D-STED. Based on this, I assume that 2D-STED has been used and for my interpretation of the results.

Response:

We are sorry we missed the references, which we have now added, along with the thickness of the sections and the information that we use of the STED microscope at 2D.

Page 21 line 21 (former page 19); Page 22 line 9.

Comment Reviewer 2-4:

General comment to fluorescence representation in all figures:

- The size of the images is rather small and capturing a large sample area, making it hard if not impossible for the reader to confirm the described localization pattern in the text. The authors have reserved a lot of space for depicting graphs of proteomic data, which could also be presented in a list or a table, while the key imaging-experiments for validating the screen are squeezed into very limited space and provide no real chance for the reader to see the confirmation but only to believe it. The images should be enlarged and panels displaying multiple region of interested be shown at larger magnification (zoom-in). For example, in Fig. 1f or h (see also image attached).

Here, after zooming in on a screen one can really see that there could be indeed three lines of titin and the traces become more believable. In addition, for each image, individual channel images and then a merge should be presented. Panels where half the image is just one channel and the other half is a merge can be very misleading (see Fig. 1 f and h).

- For each multi-colour image, at least one, but better multiple line profiles should be displayed for all fluorescent markers.

Response:

As the relation of the biotinylation sites to protein domains in heart vs. skeletal muscle cannot easily be communicated in a table, we made an extra effort to visualize the proteomics data for each protein. We have now additionally included the data in tables in the requested source data file, which illustrates the difference.

We followed the suggestions and have now increased the size of our display items, which in addition to the traces should make the localization of biotinylated proteins and antigens more obvious.

We now display line markers with mean +/- SEM for 9 traces each.

Page 19-20 and Figures 1+2 (Page 27-29), Extended data figure 2.

Comment Reviewer 2-5:

- The authors should indicate in the materials and methods section, which fluorophores correspond to which colour code in the figures (e.g. if Star 580 is always encoded in green).

Response:

In the revised manuscript we consistently display Abberior STAR 635P signal in red, Abberior STAR 580 in green.

We have added this information on Page 22, line 19ff

Comment Reviewer 2-6:

- Stating partial co-localization can be a dangerous route. Therefore, it is very important to provide the reader with all details and control experiments. The authors should also consider that the interaction could be transient or the protein be biotinylated by the non-integrated titin pool and thereby no co-localization would be detectable in immunofluorescence. The current presentation of the imaging data does neither confirm nor exclude partial-co-localization of Myosin / MyH8, Pgam2 and Neb1 (see also per figure comments below).

Response:

This is an important point and one of the messages of the manuscript – as indicated in the discussion on page 10, the BioID approach allows us to document transient colocalization as we integrate information over time and that are not easily documented by co-staining and on Page 11 we acknowledge the contribution of non-integrated titin. Our immunofluorescence images show that we find proteins identified by BioID approach in the general area of the Z-disc, although for most not exclusively. Difficulties with immunofluorescence versus BioID to determine colocalization are exemplified with Myosin, where we separate the biotinylation and the Myosin signal in the sarcomere upon stretch (no colocalization), but where collapsed sarcomeres would have shortened A-bands so that we obtain partial colocalization. Similarly, Neb1 localizes exclusively at the Z-disc but only because the available antibody labels the very C-terminus. An antibody further N-terminal would not have indicated colocalization. Thus, we include additional information to support the validity of our dataset and compare the enriched proteins with published sarcomere interacting proteins (Figure 4).

More detailed information is provided below in response to additional specific concerns – including the response to comment 2-12 with the reference to our extended discussion on Page 11, line 16-24.

Comment Reviewer 2-7a, b, c:

A more extensive (semi-) quantification of the signals (e.g. by averaging multiple line profile on different samples and displaying SD or SEM) would also improve the manuscript.

Response: We now display the traces as mean +/- SEM.

Detailed comments per figure:

- Fig. 1 f-i: Line Profiles should be shown for both colours (see also above). Both channels should be first displayed separately and then with the merge.

Response: We now display 1 f-i with separated and merged colors and have added the line profiles for all colors.

- Fig. 2 h/j: The authors should adhere to the general practice to draw line profiles in a consistent manner. This is usually perpendicular to the structure of interest, here the biotinylation region.

Response: We have harmonized the display items and provide the traces added the line profiles as requested.

Page 27-29 (Figures 1+2), Extended data

Comment Reviewer 2-8:

- Ext. Data Fig. 2: k-o partial co-localization is hard to confirm by the reader due to the small size of the images and lack of line profiles (see general comments above). Furthermore, in k, the Nebulette signal at the sarcolemma is saturated and preventing to estimate the fraction of protein at the z-disk. The authors might want to clarify this by showing non-normalized line profiles.

Response:

We have revised the display items to make them larger and picked a different nebulette image with less signal around the membrane and added the requested traces compatible to the main body.

Extended Data Page 3, Figure 2

Comment Reviewer 2-9:

Signal crosstalk between the STED dyes used in this study has been observed on other occasions when the signal intensity was substantially stronger in one channel than the other. The authors should briefly clarify the used fluorophore combination and add the respective controls to the figure (e.g. by presenting representative single staining of the structures imaged with the same microscope settings).

Response:

The filters in our STED setup separate the dyes quite well. We have added examples for single staining as well as controls without 1st antibody to the source data file.

Comment Reviewer 2-10:

In my opinion, the authors have undertaken a great task of localising sarcomeric proteins with advanced light microscopy, and this can be really helpful for the field. To establish precisely the co-localization pattern and degree of overlap (and make this easily visible to all readers), images displaying the biotinylation band and a immunostaining for a protein that resides

- a) Exclusively outside the z-disc
 - b) Exclusively at the z-disk and
 - c) are present (to varying degree) in both regions
- should be added.

A relative “sarcomeric map” should be made (see also Kelu et al., Dev Biol, 2017, <https://doi.org/10.1016/j.ydbio.2017.03.031>; Figure 9) to support the reader with the relative position of any protein that was measured.

Response:

This information is in part included in the proteomics data figure 4 (compare legend Z-disc, A-band, I-band, M-band) and we have added the requested graph to the main body of the manuscript (similar to Kelu et al Dev Biol 2017).

Page 29, Figure 2h

The staining with a protein outside the Z-disc is myosin in the stretched sarcomere (Extended Data Figure 2m), a protein exclusively at the Z-disc is Neb1 (Extended Data Figure 2k) and a protein more distributed is Pgam (Extended Data Figure 2l).

Comment Reviewer 2-11:

Although, this work has the full potential to become pioneering work. The currently included experiments and especially the data presentation provokes the feeling of being partially unfinished and causes unnecessary ambiguity. In my opinion, additional experiments could strengthen, the overall message of this work and I would like to urge the authors to consider including the following experimental methodologies:

1) To support the biochemical data further, a super-resolution immune-fluorescence mapping of titin structure by employing antibody directed against various Ig domains of titin (that can possibly be resolved by STED) would be more than supportive for the findings presented. This could establish a better overview of the confirmation of titin in stretched and non-stretched muscle and provide additional evidence for the biotinylation pattern of titin.

Response:

Please compare comments 2-6 and 2-11. In addition, there are no titin Ig-Antibodies that would specifically resolve titin at the edge of the Z-disc. The biotinylation pattern is provided in the proteomics data and has been replicated – not only technically but also in two types of muscle. The identity of titin as the biotinylated protein results from the amount of label identified by MS. Based on the proteomics data, there are only two regions of titin that provide sufficient signal to explain the presence of the two main biotinylation sites. This is now indicated in the new overview in Figure 2h, and we have used a schematic drawing to provide the biotinylation data not only in relation to the titin protein (Figure 1e) but also in relation to the sarcomere structure (Extended data Figure 5).

Comment Reviewer 2-12:

2) A mapping of myosin in respect to the biotinylation band using different antibodies recognizing the rod, the globular domain and the light chains in non-stretched and in stretched muscle should provide a detailed view to what extent the myosin is indeed coming close or penetrating the z-disc. I can imagine

that antibodies recognizing the light chain or the globular domain could provide a better and more convincing overlap with the biotinylation band. In addition, this would remove ambiguity that the biotinylation band found ~200nm away from the z-disc in the stretched muscle is formed by other proteins than myosin.

Response:

We are sorry that we did not well explain the nature of the peaks in stretched muscle. The signal flanking the Z-disc is the area, where the BioID resides in resting muscle (not stretched or contracted). In stretched muscle, some of the titin proteins are extended so that their biotinylated region is partially pulled from the Z-disc. The signal at 200nm is not myosin, but titin. As indicated in the staining of myosin and biotin in figure 2k the A/I junction (the very tip of the myosin filament) is another >100 μ m away from the outside biotin signal, which is consistent with the biotinylated myosin signal in the proteomics analysis.

The distance between two myosin heads is 14.5 nm. With an Antibody at 15nm wide it would not be possible to resolve substructures of head vs. light chain.

We have revised the figures to add additional labels for the myosin border where we only find a minor biotin peak. We revised the results and discussion explain the distribution of biotin and resulting hypotheses more clearly (compare response to reviewer 3).

Page 7 top, Page 11, line 16-24.

Comment Reviewer 2-14:

3) Single muscle fibres have been shown to provide high quality STED images. The authors might want to extend their study using these kind of samples.

<https://www.ahajournals.org/doi/pdf/10.1161/CIRCRESAHA.112.274530>

Response:

We optimized procedure quite a bit to provide cardiomyocyte stainings of suitable quality and revised the figures with higher magnification and independent samples to further improve quality. Unfortunately the quality of the stainings is limited by the sample material and also by the antibody used.

For contraction of the myocytes, we had to use skinning and Calcium, this makes it more challenging to visualize the sarcomere at even shorter sarcomere lengths. We have obtained proper traces, which confirm the localization of the biotinylation site and increased localization of the edge of the myosin filament at the Z-disc with shorter sarcomere lengths.

Page 27 figure 1; page 29, figure 2, extended data set figure 2n.

Comment Reviewer 2-15:

Minor comments but important aspects of the manuscript that can be changed for the ease of understanding of a broader audience or non-titin specialist skeletal muscle biologists:

1) Throughout the manuscript, the supplementary figures should be named as “supplementary” not as “extend figures” or can be presented the other way around but consistently

Response:

Thank you for pointing this out. We have changed this throughout the manuscript.

Comment Reviewer 2-16:

2) Page 5 it should be traces 1g vs i NOT 1e vs i

Response:

Thanks for spotting this- we changed the labels according to the revised figure in f vs. g.

Comment Reviewer 2-17:

3) Please mention if an image is confocal, see Fig 1c

Response:

All figures that are not STED are confocal. We added this to the manuscript.

Page 22 line 19-21

Comment Reviewer 2-18:

4) Page 6, the authors reported to have found “14” biotinylated proteins in the main text and “15” in the extended Data Fig.2 legend.

Response:

We are sorry that this escaped our proofreading. We found 15 proteins as indicated in figure 2a. We changed 14 to 15 on Page 6, line 4.

Comment Reviewer 2-19:

5) Supplementary Table 2: Neonatal Quad vs Neonatal Heart, there is a significant number of collagen proteins missing from Neonatal Quad, even though collagens have been shown to be essential for skeletal muscle development. This should be discussed and what could this mean in terms of the developmental differences between the two muscle types?

Response:

We consider collagen as background since it is extracellular and titin is intracellular. Alternatively, there are small amounts of extracellular titin (thus titin has been found in the urine – suggested as a novel biomarker for muscle disease). The amounts of collagen detected here could relate to the amount of collagen and also to the amount of titin shed from the cell, which would make it difficult to draw conclusions on the role of collagens in development. Indeed the role of different collagens in muscle development has not sufficiently been addressed in the literature so far. It makes for an interesting question that would need to be resolved to with a more global quantitative proteomics approach (not limited to the titin neighborhood). In this manuscript we would like to focus on the intracellular myofilament to not add an additional level of complexity.

We have added this information to the revised manuscript on page 7 line 25 to page 8 line 2.

Comment Reviewer 2-20:

6) Page 10: Second paragraph “As we integrate colocalization information over time”: it is not necessarily clear if author have done any time lapse microscopy or want to refer to their biotinylation approach.

Marko Lampe with substantial support by
Muzamil Majid Khan

Response:

We are sorry we did not explain that well. We did not do time-lapse microscopy, but refer to the continuous labeling of proximal proteins by the BioID constantly provided with biotin. This has been added to the manuscript.

Page 10 line 10-11; Page 19, 17-19.

Reviewer #3 (Remarks to the Author):

Comment Reviewer 3-1:

The authors present a very interesting approach for testing protein-protein interactions in the cardiac and skeletal muscle sarcomere. By generating a BioID mouse with a biotin-ligase located in the Z-disk region of titin the authors are now able to monitor titin interactions with highly improved spatio-temporal resolution, compared to previous approaches.

The experiments are well-performed, statistical analyses are appropriate, description of the methods is short but precise.

Key findings of the paper are:

- identification of 478 sarcomeric or sarcomere associated proteins, which is much more than suggested before.
- identification of titin Ig8/9 as putative hairpin forming structures that may influence titin-based elastic properties of the sarcomere
- confirmation of previous hypotheses suggesting Z-disk penetration by myosin filaments during contraction/relaxation cycle
- Description of titin as a dynamic signaling hub for signal transduction and metabolism, depending on developmental state.

The manuscript is packed with excellent original data and provides extensive supplementary material. Some of the data are confirmatory, others really novel and exciting.

Response:

Thank you for the kind assessment and the appreciation of our work.

Comment Reviewer 3-2:

Description and interpretation of these large data sets in a comprehensive manner and for a large readership is indeed challenging. The reader is overwhelmed by the different data that range from general protein interaction studies, structural titin-spring studies to developmental changes in metabolism. Not to mention the data from proteome analyses.

At some points the selection of data shown in the graphs seems random, e.g. Figure 2. Here, biotinylated metabolic enzymes are presented, including Ldb3, Phtf1 and Pgam2, but most of the proteins shown are not really discussed later in the manuscript.

Response:

We agree that the proteomics approach provides a large amount of data, which benefits from additional care in downstream analysis. Most critically when we have a limited subset of biotinylated proteins as identified in the antibody pulldown, it is necessary to properly document biotinylation for all of them. The figures were reassembled based on the following goal: (1) to show that skeletal and heart biotinylation sites overlap, providing additional validation of the experimental approach with an

independent replicate. (2) to display whether biotinylation is restricted to certain domains or unstructured regions. (3) to provide a visual representation of the amount of biotinylation within proteins.

We reshuffled Extended Figure 2c-j and now include each biotinylated protein separately.

We also added an overview to the main body together with an overview that summarizes the staining results (compare reviewer 2). All remaining stainings are in the supplement.

Page 6 line 6 ff, Figure 2, Extended Data Figure 2

Comment Reviewer 3-3:

For the reader it is very hard to follow all these different lines of evidences without getting lost in details of the figures. One suggestion could therefore be to focus the manuscript body to the main findings and move some of the detailed graphs to the supplement or to an independent manuscript.

Maybe then it would be possible to go into more detail when explaining e.g. the titin hairpin theory. I think this is really exciting!!

Response:

To improve the understanding of display items representing a biotinylated protein, we now refer to them individually in the results section.

In the main body, we were aiming to provide an example for each type of protein (e.g. expected, structural, metabolic, ER). In addition, we decided to highlight proteins with strong biotinylation and focus on skeletal vs. cardiac myosins based on the more orderly arrangement of skeletal myofilaments for better quality immunofluorescence pictures (Figure 2 vs. Extended Data Figure 2).

We have rearranged supplementary figure 2 to better structure the omics and imaging data and extended the description of the hairpin theory.

Page 10 line 22-25

Comment Reviewer 3-4:

I was a bit lost when looking for the additional subset of identified SR-titin interactions, mentioned on page 7. This could also be described in more detail.

Response:

We have revised the section on page 6.

Page 6, line 6-16

Comment Reviewer 3-5:

I also feel that a few sentences on the principle of integration of a BioID should be added to the introductory section.

Response:

We added this information to the introduction and extended results.

Page 3, line 18&19, Page 4, line 5&6

Comment Reviewer 3-6:

However, the discussion is really well structured and very well written and thus helps understanding the different findings of the manuscript.

Martina Krüger

Response:

Thank you - we hope we have now amended the results section to the same level as the discussion.

REVIEWERS' COMMENTS:

Reviewer #1 (Remarks to the Author):

The author have addressed my concerns, although I am confused as to the information provided in the Excel file titled fig 3a, fig 3b, fig 5a, fig 5b. These appear to be iBAQ values, but there are no protein identifiers for the values.

Reviewer #2 (Remarks to the Author):

The authors of Rudolph et al. "Deconstructing sarcomeric structure/function relations in titin-BioID knock-in mice" have substantially improved their manuscript by adding and amending text and figures. The imaging part can now be judged and understood by the reader. All essential information is now present in the manuscript and the accompanying extended material.

A few minor points should be addressed before publication:

- Text page 7 line 6: "Fig 2h,i" should read "Fig2i,j" and "Fig 2j,k" should read "Fig 2k,l"

- Fig2i and Fig2k: The line profile indicated in the image is parallel to the border of the figure. It is standard practice to measure the line profile perpendicular to the structure of interest (please also see image attached to this letter) and this should be corrected before publication. Fig 2j and Fig 2l should be updated accordingly.

In addition: The authors might want to consider improving readability of protein names/abbreviations in Fig4 and Ext. Fig 3d.

We hope that the interesting findings regarding the force generating mechanism in muscle will be followed up in future work and extend this elegant study.

Marko Lampe

with substantial support by

Muzamil Majid Khan

Reviewer #1 (Remarks to the Author):

Comment Reviewer 1-1:

The author have addressed my concerns, although I am confused as to the information provided in the Excel file titled fig 3a, fig 3b, fig 5a, fig 5b. These appear to be iBAQ values, but there are no protein identifiers for the values.

Response:

Thank you for noticing – we have added the protein names for all sheets in question (consolidated in the new *Source Data.xlsx*).

Reviewer #2 (Remarks to the Author):

Comment Reviewer 2-1:

The authors of Rudolph et al. “Deconstructing sarcomeric structure/function relations in titin-BioID knock-in mice” have substantially improved their manuscript by adding and amending text and figures. The imaging part can now be judged and understood by the reader. All essential information is now present in the manuscript and the accompanying extended material.

A few minor points should be addressed before publication:

- Text page 7 line 6: “Fig 2h,i” should read “Fig2i,j” and “Fig 2j,k” should read “Fig 2k,l”

Response:

Thanks for pointing this out. We have changed the text to accommodate the insertion of figure 2h in the first revision.

Comment Reviewer 2-2:

- Fig2i and Fig2k: The line profile indicated in the image is parallel to the border of the figure. It is standard practice to measure the line profile perpendicular to the structure of interest (please also see image attached to this letter) and this should be corrected before publication. Fig 2j and Fig 2l should be updated accordingly.

Response:

We have updated figure 2j and l as suggested.

Comment Reviewer 2-3:

In addition: The authors might want to consider improving readability of protein names/abbreviations in Fig4 and Ext. Fig 3d.

Response:

These are indeed data-heavy figures. We have added a white outline to the protein/names and abbreviations and realigned them to improve readability.

Comment Reviewer 2-4:

We hope that the interesting findings regarding the force generating mechanism in muscle will be followed up in future work and extend this elegant study.

Marko Lampe
with substantial support by
Muzamil Majid Khan

Response:

We thank the reviewer for the encouraging words and help to improve the manuscript.